REGISTERED REPORT PROTOCOL

# Identifying meta-research with researchers as study subjects: Protocol for a scoping review

Gregory Laynor[1]*, Elizabeth R. Stevens[2]

**1** NYU Health Sciences Library, NYU Grossman School of Medicine, New York, NY, United States of America, **2** Department of Population Health, NYU Grossman School of Medicine, New York, NY, United States of America

* Gregory.Laynor@nyulangone.org

This is a Registered Report and may have an associated publication; please check the article page on the journal site for any related articles.

**Data Availability Statement:** All relevant data are within the paper and its Supporting Information

## Abstract

### Background

Meta-research in which researchers are the study subjects can illuminate how to better support researchers and enhance the development of research capacity. Comprehensively compiling the literature in this area can help define best practices for research capacity development and reveal gaps in the literature. However, there are challenges to assessing and synthesizing the breadth of the meta-research literature produced.

### Methods

In this article, we discuss the current barriers to conducting literature reviews on meta-research and strategies to address these barriers. We then outline proposed methods for conducting a scoping review on meta-research with researchers as study subjects.

### Discussion

Due to its interdisciplinary nature, broad scope, and difficult to pinpoint terminology, little is known about the state of meta-research with researchers as the study subjects. For this reason, there is a need for a scoping review that will identify research performed in which researchers were the study subjects.

## Background

Scoping reviews play an important role in the development and improvement of research questions and strategies. They allow researchers to assess the current state of research on a topic, determine who experts are, identify key questions that need further research, and ascertain methodologies used in past studies [1–3]. Specifically in the field of meta-research (research on research), scoping reviews have been conducted on topics such as research waste [4], the use of systematic reviews in planning new studies of health research [5], guidelines for creating biomedical journals [6], the tasks of peer reviewers [7], clinician-researcher careers [8], and the training of clinician-scientists [9].

files. Specifically, we are including as a Supporting Information File the data from the testing of the preliminary search strategy. We intend for this Supporting Information File to be made openly available as part of the publication of the paper.

**Funding:** Funding was received from National Institute on Aging (NIA) for this work (Stevens - 1K01AG075169-01A1). The funders did not and will not have a role in study design, data collection and analysis, decision to publish, or preparation of the manuscript.

**Competing interests:** The authors have declared that no competing interests exist.

To capture literature relevant for answering a research question, identifying an appropriate set of search terms is a critical step in the development of a literature search [10,11]. Defining search terms, however, can pose a particular challenge depending on whether a topic has differing terminology in multiple fields, terminology that has changed over time, or a limited maturity of a research field leading to less well characterized key terms [12,13].

The field of meta-research is one such area that lacks a well delimited set of key terms [14]. As a discipline designed to study research itself and its practices, meta-research is not only multidisciplinary, but is also comparably new [15,16]. Having been coined less than a decade ago [17] , meta-research is generally lacking widespread recognition or a united set of terminology. Although newly defined, researchers have been performing meta-research for decades on a number of topics and in a variety of fields, but rarely under the label of "meta-research." In part due to the relatively recent coalescence of this type of research into a discipline, the term meta-research has not been consistently used and literature databases do not have established key indexing terms for this type of research (e.g. PubMed Medical Subject Headings (MeSH) terms). To add to the confusion, the term meta-research is also occasionally used to refer to meta-analysis, which is a research process used to systematically synthesize the findings of independent studies and distinct from meta-research the discipline.

Given the broad scope of meta-research [18], it can be challenging to retroactively define a unifying set of key terms, particularly as many of the topics explored and methods used in meta-research overlap with common terms and phrases used in most other research fields. Some meta-research themes lend themselves more readily to easily distinguishable key terms such as "research ethics," "reproducibility," or "peer review" that can be used in a systematic literature review to identify research in these categories. Other meta-research topics, however, have less straight forward keywords, as many of the key terms used in the discussion and methods overlap with those used in other topics and/or are commonly used words.

One such meta-research topic without easily identified search terms includes research falling within the theme of incentives (how research is rewarded or supported, e.g. promotion criteria, developing research capacity) [18]. This research often has researchers as the study subjects and can be used to assess needs, behaviors, and perceptions among researchers. Just as the examination of research practices in other areas of meta-research can help disseminate efficient and effective research policies and to identify and abandon wasteful ones, insights from the body of literature with researchers as the study subjects can illuminate how to better support researchers, enhance the development of research capacity, and advance the methods used to engage and study this population [15]. However, it can be a challenge to assess and synthesize the breadth of the meta-research literature produced. For instance, one term for the population studied, "researcher," is used in tens of millions of published articles and therefore a less useful term for identifying a study population than the term "smoker," for example, which is found in less than one-fourteenth the number of articles.

Due to its interdisciplinary nature, broad scope, and difficult to pinpoint terminology, little is known about the state of meta-research with researchers as the study subjects. Comprehensively compiling the literature in this area can increase the visibility of the research being performed, help define best practices for research capacity development, and provide insight into topics that require further investigation. For this reason, there is a need for a scoping review that will identify research performed in which researchers were the study subjects.

A scoping review on researchers as study subjects will initially be focused on health-related disciplines, as a feasible starting point for this potentially vast area of research, Future scoping reviews may expand the scope beyond health-related disciplines. The scoping review will establish a bibliometric summary of where this literature has been published, reveal gaps in the literature, and define the scope of research in this area, as well as synthesize any "lessons

learned" for engaging with this study population. In this article, we present a strategy for addressing the current barriers to conducting literature reviews on meta-research and outline methods for conducting the proposed scoping review on researchers as study subjects.

## Methods

The scoping review protocol is guided by the methodological frameworks established by the Joanna Briggs Institute (JBI) scoping review methodology and by Arksey and O' Malley (2005) [2,19]. The review process will also be reported in accordance with the Preferred Reporting Items for Systematic reviews and Meta-Analysis extension for Scoping Reviews (PRISMA-ScR) reporting guidelines (S1 Table) [20].

### Stage 1: Identifying the research question

The review aims to answer the question: What research has been performed where those performing research are the study population? Our objectives are to: First, describe what research has been performed where researchers are the study subjects; second, identify the data sources and methodologies used when researching these populations; and third summarize the identified barriers to performing research within these populations.

### Stage 2: Identifying relevant studies

Data sources for the scoping review will be collected from across health-related disciplines. The search strategy will focus on three core databases in health-related disciplines: PubMed, CINAHL, and PsycInfo. The search strategy will thus yield both peer reviewed published literature in electronic databases, as well as two databases (CINAHL and PsycInfo) that also index grey literature including theses, dissertations, and conference proceedings. For the purpose of this review, a researcher is defined as anyone engaged in conducting academic or scientific research regardless of discipline, topical focus, or stage of career (e.g. professor vs. research assistant).

**Eligibility criteria.** To be included in the review a source must: (1) have an English language abstract (2) include researchers as the subjects of the research; (3) not focus solely on the engagement of community members in research (e.g. Community Based Participatory Research [CBPR]) or physicians who are subjects in quality improvement research. The subjects of quality improvement research are presumed to be actors in the health practice and not performing the research themselves. A study of physicians performing quality improvement research themselves, however, would be included.

**Search scope.** The scope of this literature review was restricted to fields within the domains of health and well-being. This decision was made for both pragmatic and theoretical reasons. Pragmatically, limiting the review scope made the process of screening for relevant articles more feasible. Given the broad scope of the review and limited available specific search terms, working within one domain allows for a less specific and more sensitive search without making the screening process untenable. One goal of this review is to determine an effective search strategy to help define the subject area. Once defined in the health field, translating the strategy to other fields will be more attainable. Furthermore, a second goal of this review is to summarize the current state of the literature. While it is anticipated that many of the same issues will impact researchers across fields, in theory, researchers in different fields are likely to have some issues that are unique to those in that fields. Therefore, limiting the review to research pertaining to health researchers will promote the ability to be more comprehensive summarizing the literature and therefore facilitate identification of aspects of research-on-researchers that are specific to those in the domain of health research. The methods outlined

in this review could be adapted to other fields beyond health research and could be expanded to include further databases to ensure a comprehensive search [21].

**Search strategy.** The search strategy aims to capture research performed on researchers in health-related domains. To this end, three core disciplinary databases will be searched: PubMed, CINAHL, and PsycInfo. In addition, known relevant literature and other literature not found via the database search will be explored by "snowballing" the references of the items identified in the database search [22].

The development of the search strategy began with harvesting keywords as well as terms from PubMed's structured vocabulary, Medical Subject Headings (MeSH), from a set of studies already known by the researchers that would meet the eligibility criteria for study inclusion established in the protocol (S2 Table). The strategy was subsequently tested for retrieval of the known studies, with iterative expansion and refinement of the search strategy.

The topic of this scoping review presents specific challenges for search strategy development, in that key terms are frequently used across research areas, particularly the term "researcher" and its synonyms. The broad use of the term "researcher" in the literature limits its functionality as a keyword yielding an untenable number of documents. Therefore, "researcher," and its identified synonyms, were restricted as keywords to the title and abstract based on the assumption that most literature with researchers as the study population will include these terms prominently. This strategy, however, still yielded a large number of results and the majority of the results would be expected to be eliminated during the screening process. Due to these issues, an alternative strategy for developing the search was taken that utilizes available MeSH terms as well as phrases searched with proximity searching.

The search strategy, initially developed in PubMed and then translated to the other databases, combines two approaches. The first approach utilizes two components:

Component A: One component of the search names terms for researchers, in addition to utilizing the MeSH term for "Research Personnel," these keywords are included: clinician-scientist; investigator; physician-scientist; professor; project team; research assistant; research faculty; research personnel; research staff; research team; research worker; research workforce; researcher; scientist; study personnel; study staff; study team; trialist.

Component B: These names are then combined with a series of keywords identifying actions performed by researchers as study subjects: engaged; engagement; engaging; experiences; involved; involvement; participate; participated; participating; participation; perceptions; responded; responding; responses.

Each term from Component A for naming researchers, in addition to plural and possessive spellings where appropriate, are combined with each term from the Component B in phrases searched with proximity searching of two words. For example, the phrase "clinician-scientist engaged" searched with a proximity operator of two means that clinician-scientist and engaged must appear within two words of each other in a title or abstract of an article to be included in the search. This proximity operator improves the specificity of the search [23]. By combining in phrases each term from Component A with each term from Component B, the search ensures sensitivity.

This search is also supplemented by a second approach, to further identifying a smaller subset of studies naming meta-research or research-on-research. This was done by combining the terms from Component A (using the Boolean operator OR within the set) with the Boolean operator AND with a set of terms describing meta-research, meta-science, research-on-research, and research-on-researchers (by including the most subheadings of the MeSH term for "Research Personnel" found in the search development to yield the most relevant results to research on researchers as study subjects: "psychology" (to capture studies on perceptions or thought processes of researchers), "trends"; and "statistics and numerical data." The two

approaches are them combined with the Boolean operator OR, so that studies from either or both of the approaches are included in the search results.

The search was tested on a preliminary set of known studies to ensure high yield. The core PubMed search strategy, as well as the testing of the search (with the yield number) are included in S3 Table. This search strategy will be adapted for each database, combining both approaches (proximity searching from iterated phrases and then a more focused search on terms of meta-research). For each database, any available subject headings from the controlled vocabulary of each database will be used.

In addition to the searches of PubMed, CINAHL, and PsycInfo, subsequent to the screening process, we will also conduct forward and backward citation chasing, "snowballing," of included articles. The initial searches, combined with chasing citation of included studies, aims to provide a comprehensive and sensitive search of publications on researchers as study subjects, without creating an untenable number of citations to screen due to challenges in this topic's terminology as discussed in this protocol article.

## Stage 3: Study selection

After removing duplicate references in Covidence, the selection of included literature will occur in two screening phases. In the first phase two reviewers will independently screen each title and abstract using a screening protocol based on the identified eligibility criteria. To minimize the risk of bias, the first phase screening protocol will be pilot tested by reviewers on a random selection of 100 titles and abstracts to ensure consistency and reliability. Literature identified as potentially relevant will be passed to the next screening level. In the second phase, two reviewers will independently review full-text versions of all potentially relevant literature. Reviewers will be trained on the phase two screening protocol prior to beginning screening and the protocol will be pilot tested by the reviewers on a random selection of 10 full texts to ensure consistency and reliability between the reviewers.

In both phases, the inter-investigator agreement will be quantified with a kappa cut off greater than 0.8 indicating almost perfect agreement [20]. Disagreements will be resolved to consensus through discussion. Any unresolved disagreements will be referred to a third investigator for review. A PRISMA flow diagram will be created to report screening result numbers once the review is complete. Covidence will be used to manage the search results.

## Stage 4: Charting the data

Two reviewers will independently read each included article. Using the data extraction form created for this review, one reviewer will extract relevant data using a standardized data extraction tool, and another reviewer will verify the extracted data for consistency and accuracy. Any inconsistencies will be resolved through discussion. The draft extraction tool will be piloted with five studies and modify as needed. The data extracted for this review will include: author (s); year of publication; country of origin; funding source; study aim/purpose; study population and sample size; discipline setting; study design; data collection methods and types of data used; response rates; barriers and facilitators to research on researchers. This process will be iterative, and the extraction form will be updated as new elements are added.

## Stage 5: Collating, summarizing and reporting results

A tabular report will be produced summarizing the extracted data aligning with the objectives and scoping review questions. We will conduct an inductive thematic analysis on the qualitative data (i.e., barriers and facilitators). Studies will be mapped according to their contextual setting, geographical location, and year of publication. Finally, a narrative summary will describe the

findings and how they relate to the review's research question and objectives. These results will identify gaps in the literature and potential strategies for performing research with researchers as the study subjects. The summaries will provide a bibliometric summary of how much research has been performed on researchers, what topics are being explored, the type of data being used, and the barriers and facilitators to studying researchers including response rates.

## Conclusion

There is a need for a better understanding of the research landscape of this area of meta-research, research-on-researchers. Scoping review methods provide a means to develop this understanding. Due to the terms of this area of meta-research being so common across much of the (largely unrelated) research landscape, developing a scoping review search strategy for this topic is more difficult than typical search strategy development, even in some areas of meta-research where scoping reviews have been conducted on more circumscribed domains such as publishing practices. To address the challenge of identifying meta-research on researchers as study subjects (research-on-researchers), iterated phrases and proximity searching were used, after a process of testing different approaches to the search strategy on a sample set of known studies that would fit eligibility criteria for inclusion in the scoping review. While the search was developed to balance sensitivity and specificity, the study still has limitations. One way of addressing these limitations, without increasing search sensitivity to the point of having citation sets for screening beyond what is feasible to screen while also having likely low yield for inclusion, is the use of iterated phrases from key components of the topic searched with proximity searching. Such an approach may be useful for other scoping reviews on topics that are difficult to search due to having terminology that is also widely used across the research landscape in unrelated fields and topics.

This scoping review will provide a basis for further studies on researchers as study subjects, as well as providing a foundation for defining research-on-researchers as a sub-field of meta-research. Because this review will only search citation databases in health-related fields, this study will only look at a subset of researchers as study subjects. The methods could be expanded to study researchers as research subjects in other fields. Additionally, this study provides a model for developing search strategies on topics difficult to search due to their interdisciplinarity, broad scope, or lack of well-defined terminology.

## Supporting information

**S1 Table. PRISMA checklist.**
(DOCX)

**S2 Table. Testing of search strategy sensitivity on initial set of eligible studies.**
(DOCX)

**S3 Table. Core search strategy (PubMed).**
(DOCX)

## Author Contributions

**Conceptualization:** Gregory Laynor, Elizabeth R. Stevens.

**Funding acquisition:** Elizabeth R. Stevens.

**Methodology:** Gregory Laynor, Elizabeth R. Stevens.

**Writing – original draft:** Gregory Laynor, Elizabeth R. Stevens.

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
