## [Decision Letter · Decision Letter 0]

9 Jan 2024

PONE-D-23-25128Identifying meta-research with researchers as study subjects: protocol for a scoping reviewPLOS ONE

Dear Dr. Laynor,

Thank you for submitting your manuscript to PLOS ONE. After careful consideration, we feel that it has merit but does not fully meet PLOS ONE’s publication criteria as it currently stands. Therefore, we invite you to submit a revised version of the manuscript that addresses the points raised during the review process.

We look forward to receiving your revised manuscript.

Kind regards,

Lutz Bornmann

Academic Editor

PLOS ONE

Journal Requirements:

2. In your cover letter, please confirm that the research you have described in your manuscript, including participant recruitment, data collection, modification, or processing, has not started and will not start until after your paper has been accepted to the journal (assuming data need to be collected or participants recruited specifically for your study). In order to proceed with your submission, you must provide confirmation.

3. Please expand the acronym “NIA” (as indicated in your financial disclosure) so that it states the name of your funders in full. 

"Funding: Funding was received from NIA for this work (Stevens - 1K01AG075169-01A1). The funder had no role in  the preparation of this study.

 Conflicts of Interest: The authors have no conflicts of interest to declare.

 Author Contributions: ERS conceived of the study, contributed to methods development, acquired funding, and   drafted the manuscript. GL conceived of the study, contributed to methods development and contributed to    manuscript writing   and is guarantor of the review."

Funding information should not appear in the Acknowledgments section or other areas of your manuscript. We will only publish funding information present in the Funding Statement section of the online submission form. 

"ES received funding from the NIA (https://www.nia.nih.gov/) (Grant--1K01AG075169-01A1). The funders did not  and will not have a role in study design, data collection and analysis, decision to publish, or preparation  of the manuscript."

6. Please upload copies of the completed PRISMA checklist as Supporting Information with a file name “PRISMA checklist”.

**Additional Editor Comments:**

The second reviewer has not seen that you submitted a Registered Report Protocol. Thus, please ignore his/her corresponding comments. 

Reviewers' comments:

Reviewer's Responses to Questions

**Comments to the Author**

1. Does the manuscript provide a valid rationale for the proposed study, with clearly identified and justified research questions?

Reviewer #1: Yes

Reviewer #2: Partly

2. Is the protocol technically sound and planned in a manner that will lead to a meaningful outcome and allow testing the stated hypotheses?

Reviewer #1: Yes

Reviewer #2: No

3. Is the methodology feasible and described in sufficient detail to allow the work to be replicable?

Reviewer #1: Yes

Reviewer #2: Yes

4. Have the authors described where all data underlying the findings will be made available when the study is complete?

Reviewer #1: No

Reviewer #2: No

5. Is the manuscript presented in an intelligible fashion and written in standard English?

Reviewer #1: Yes

Reviewer #2: Yes

6. Review Comments to the Author

You may also provide optional suggestions and comments to authors that they might find helpful in planning their study.

Reviewer #1: The manuscript is a protocol for data collection for a scoping review on studies of researchers as study subjects and outlines why searching for the literature for studies concerned with researchers is difficult. This is an important concern for scoping reviews on meta-research as the terminology is very heterogenous and the term 'researcher(s)' itself very common and unspecific.

For the study being preregistered, the authors focus on research in the health scieces in order to limit the scope of the potentially vast literature. A search strategy using a series two-component combination phrases is proposed, the components being terms for researchers and terms for their actions. This was supplemented by also searching for the combination of the first component with terms relating to meta-research.

Minor remark.

p. 12 l. 280 a word seems to be missing in this phrase: "studies that would eligibility criteria for inclusion in the scoping review."

I think this protocol is ready for publication.

Reviewer #2: The manuscript is about meta-research with researchers as subjects. The manuscript proposes concepts for conducting meta-research, in particular a scoping review in this area. Except for examples in the appendix, no specific scoping review has been conducted on a topic.

The manuscript addresses an interesting and important topic for quantitative science studies and is written in a clear and concise manner, but it cannot be recommended for publication for the following reasons.

- Criteria for publication: The manuscript is a proposal and does not report the results of an empirical study. According to the criteria for publication (https://journals.plos.org/plosone/s/criteria-for-publication), “hypothesis or proposal papers” should not be published in PlosOne.

- Literature: The manuscript does not reflect literature from “Quantitative Science Studies”, “Science of Science”, or “Research on Research”, which are mainly concerned with this topic, also in meta-perspective. The revised manuscript should demonstrate the need for meta-research from the existing literature.

- Interdisciplinarity: Researchers as objects of study can be examined from very different perspectives, such as psychological (e.g. motivation), medical (e.g. influence of testosterone), statistical and sociological. I wonder whether scoping reviews of this kind do not also require special expertise in a field and special questions in order to be able to draw conclusions from them. However, this is not discussed in the draft.

- Lack of differentiation to meta-analysis: The proposed steps are more or less identical to those of a meta-analysis. I wonder if there is a big difference between scoping reviews and meta-analysis or ordinary reviews.

- Reference style: According to the “Vancouver” style, PLOS One requires the following: ”In the text, cite the reference number in square brackets (e.g., “We used the techniques developed by our colleagues [19] to analyze the data”). PLOS uses the numbered citation (citation-sequence) method and first six authors, et al.” ( https://journals.plos.org/plosone/s/submission-guidelines#loc-references).

7. PLOS authors have the option to publish the peer review history of their article (what does this mean?). If published, this will include your full peer review and any attached files.

Reviewer #1: No

Reviewer #2: No

---

## [Author Response · Author response to Decision Letter 0]

2 Feb 2024

Editor

Complete

2. In your cover letter, please confirm that the research you have described in your manuscript, including participant recruitment, data collection, modification, or processing, has not started and will not start until after your paper has been accepted to the journal (assuming data need to be collected or participants recruited specifically for your study). In order to proceed with your submission, you must provide confirmation.

Complete

3. Please expand the acronym “NIA” (as indicated in your financial disclosure) so that it states the name of your funders in full. This information should be included in your cover letter; we will change the online submission form on your behalf.

Complete

4. Please remove any funding-related text from the manuscript and let us know how you would like to update your Funding Statement. Currently, your Funding Statement reads as follows: "ES received funding from the NIA (https://www.nia.nih.gov/) (Grant--1K01AG075169-01A1). The funders did not and will not have a role in study design, data collection and analysis, decision to publish, or preparation of the manuscript." Please include your amended statements within your cover letter; we will change the online submission form on your behalf.

Complete—no changes

5. Please include captions for your Supporting Information files at the end of your manuscript, and update any in-text citations to match accordingly. 

Complete

6. Please upload copies of the completed PRISMA checklist as Supporting Information with a file name “PRISMA checklist”.

Complete

7. The second reviewer has not seen that you submitted a Registered Report Protocol. Thus, please ignore his/her corresponding comments. 

Those specific comments have been disregarded.

Reviewer #1

1. p. 12 l. 280 a word seems to be missing in this phrase: "studies that would eligibility criteria for inclusion in the scoping review."

Thank you for identifying this error. We have added to the sentence so it now reads: “…studies that would fit eligibility criteria for inclusion in the scoping review.”

Reviewer #2

We thank the reviewer for their close read of the manuscript and interest in the topic presented. As per the editor comments regarding the Registered Report Protocol, we are not addressing the comments specific to publication criteria.

1. Literature: The manuscript does not reflect literature from “Quantitative Science Studies”, “Science of Science”, or “Research on Research”, which are mainly concerned with this topic, also in meta-perspective. The revised manuscript should demonstrate the need for meta-research from the existing literature.

The first paragraph of the introduction seeks to discuss the scoping reviews that have been performed on meta-research topics (ln 60-64). We have attempted to clarify that our term of “meta-research” is research on research and have now placed clarification at the first instance of “meta-research” use. Now stating, “Specifically in the field of meta-research (research on research)…”

2. Interdisciplinarity: Researchers as objects of study can be examined from very different perspectives, such as psychological (e.g. motivation), medical (e.g. influence of testosterone), statistical and sociological. I wonder whether scoping reviews of this kind do not also require special expertise in a field and special questions in order to be able to draw conclusions from them. However, this is not discussed in the draft.

We agree that the study of researchers has the potential to be a very interdisciplinary body of literature. As a scoping review, the aim of this protocol is to create an overview of the literature available in this topic (research on researchers) including a description of the types of research—e.g. psychological, epidemiological, etc.—being performed in this population. In the protocol section “Stage 4: Charting the data” (ln 247-256) we specify the types of data to be extracted in the review and include “data collection methods and types of data used” to capture the type of research being performed. We believe that this scoping review does not require an expansively interdisciplinary to perform this summarization function. However, if future reviews are performed that include the interpretation of the results, a more interdisciplinary team will be considered. 

3. Lack of differentiation to meta-analysis: The proposed steps are more or less identical to those of a meta-analysis. I wonder if there is a big difference between scoping reviews and meta-analysis or ordinary reviews.

The search methods used in a scoping review are very similar to a meta-analysis as both seek to use rigorous and replicable search strategies, however, the type of research question being answered differs between these types of reviews. A scoping review seeks to present an overview of a body of literature pertaining to a broad topic. A systematic review (including meta-analyses) attempts to collate empirical evidence pertaining to a focused research question. We have specified in our Methods section the scoping review methods guidelines that were followed, stating: (ln 117-119) “The scoping review protocol is guided by the methodological frameworks established by the Joanna Briggs Institute (JBI) scoping review methodology and by Arksey and O' Malley (2005).”

Arksey H, O'Malley L. Scoping studies: towards a methodological framework. International Journal of Social Research Methodology. 2005;8(1):19-32.

Peters MDJ, Godfrey C, McInerney P, Khalil H, Larsen P, Marnie C, et al. Best practice guidance and reporting items for the development of scoping review protocols. JBI Evid Synth. 2022;20(4):953-68.

4. Reference style: According to the “Vancouver” style, PLOS One requires the following: “In the text, cite the reference number in square brackets (e.g., “We used the techniques developed by our colleagues [19] to analyze the data”). PLOS uses the numbered citation (citation-sequence) method and first six authors, et al.” (Link).

The formatting of the paper has been updated to reflect the style requirements of PlosOne.

---

## [Decision Letter · Decision Letter 1]

20 Mar 2024

PONE-D-23-25128R1Identifying meta-research with researchers as study subjects: protocol for a scoping reviewPLOS ONE

Dear Dr. Laynor,

Thank you for submitting your manuscript to PLOS ONE. After careful consideration, we feel that it has merit but does not fully meet PLOS ONE’s publication criteria as it currently stands. Therefore, we invite you to submit a revised version of the manuscript that addresses the points raised during the review process. Reviewer 2 has important points that should be considered.  Please submit your revised manuscript by May 04 2024 11:59PM. If you will need more time than this to complete your revisions, please reply to this message or contact the journal office at plosone@plos.org. Please include the following items when submitting your revised manuscript:A rebuttal letter that responds to each point raised by the academic editor and reviewer(s). You should upload this letter as a separate file labeled 'Response to Reviewers'.A marked-up copy of your manuscript that highlights changes made to the original version. You should upload this as a separate file labeled 'Revised Manuscript with Track Changes'.An unmarked version of your revised paper without tracked changes. You should upload this as a separate file labeled 'Manuscript'.

We look forward to receiving your revised manuscript.

Kind regards,

Lutz Bornmann

Academic Editor

PLOS ONE

Reviewers' comments:

Reviewer's Responses to Questions

**Comments to the Author**

1. Does the manuscript provide a valid rationale for the proposed study, with clearly identified and justified research questions?

Reviewer #1: Yes

Reviewer #2: Yes

2. Is the protocol technically sound and planned in a manner that will lead to a meaningful outcome and allow testing the stated hypotheses?

Reviewer #1: Yes

Reviewer #2: Partly

3. Is the methodology feasible and described in sufficient detail to allow the work to be replicable?

Reviewer #1: Yes

Reviewer #2: Yes

4. Have the authors described where all data underlying the findings will be made available when the study is complete?

Reviewer #1: Yes

Reviewer #2: No

5. Is the manuscript presented in an intelligible fashion and written in standard English?

Reviewer #1: Yes

Reviewer #2: Yes

6. Review Comments to the Author

You may also provide optional suggestions and comments to authors that they might find helpful in planning their study.

Reviewer #1: The manuscript is a protocol for the search strategy to identify studies that have scientific researchers as the studied population. This is intended as a preliminary step for a literature search for an eventual scoping review on subject of research about reserachers. The pointed out small error has been corrected. I consider the protocal ready for publication.

Reviewer #2: The manuscript is about meta-research with researchers as subjects. A registered report protocol was suggested to conduct a meta-research study on this topic with the help of bibliometrics.

The manuscript addresses a very interesting and important topic not only for the field of “quantitative science studies” and is written in a clear and concise manner. I appreciate the revision that were done to improve the manuscript The following concerns should be considered in the revision:

- Meta-research: The introduction is essentially based on a presentation of a new method, meta-research, but the real issue is the question of researchers as subjects of scientific studies and how research in this area can be synthesised. Meta-research is one of several possible methods, along with meta-analysis and systematic review, to address this question. I recommend starting with the research gap and then presenting meta-research as an approach to answering the question.

- Motivating the research question: In my view there is no sufficient derivation of the research question from the research literature (e.g., quantitative scientific studies) on questions where the researchers themselves were made the subject of the research. Reference is made to a "body of literature", but no literature is cited, although a body of empirical literature must be assumed. The question remains as to the scientific background against which the results in the registered report can be interpreted and used.

- Registered report protocol: I wonder why this study was submitted as a registered report protocol. Registered report protocols are useful if you have formulated a clear research question, hypotheses, and an experimental design to test the hypotheses. This makes it possible to publish results in the subsequent study that contradict the original hypotheses. In the study plan presented no hypotheses are formulated; it is an exploratory study. A potential risk of bias is not recognisable. The plan for the data analysis and reporting is vague (e.g., tabular reports, thematic analyses, narrative summaries). Results of a preliminary study with a bibliographic database are not reported (e.g., potential number of studies, text-mining for key words). A sufficient justification for the type of article is missing.

- Component B: It was stated (p. 9) that “These names are then combined with a series of keywords identifying actions performed by researchers as study subjects”. I wonder where these keywords (e.g., engaged, …) come from. Important verbs as for example “investigate” or “conduct” are missing. The revision should give some reasons for the selection of keywords for actions.

- Funding: The financial disclosure stated that "funding was received from the National Institute of Aging (NIA)". This raises the question of who will fund the subsequent study.

7. PLOS authors have the option to publish the peer review history of their article (what does this mean?). If published, this will include your full peer review and any attached files.

Reviewer #1: No

Reviewer #2: No

---

## [Author Response · Author response to Decision Letter 1]

29 Mar 2024

Response to Reviewers

1. Have the authors described where all data underlying the findings will be made available when the study is complete?

Reviewer #2: No

We have updated the Data Availability Statement for the study to note that we intend to make publicly available all relevant data from all stages of scoping review, including the development of the scoping review protocol and the conduct of the scoping review. Data on the testing of the preliminary search strategy will be made available as a Supporting Information file included with the manuscript. Once the scoping review is conducted, data on included and excluded citations will be included with as a Supporting Information file 

2. Meta-research: The introduction is essentially based on a presentation of a new method, meta-research, but the real issue is the question of researchers as subjects of scientific studies and how research in this area can be synthesised. Meta-research is one of several possible methods, along with meta-analysis and systematic review, to address this question. I recommend starting with the research gap and then presenting meta-research as an approach to answering the question.

Within this protocol we are using the definition of meta-research as a research discipline rather than a specific research method (i.e. the research method meta-analysis). Meta-research is a discipline designed to study research itself and its practices and the objective is to understand and improve how we perform, communicate, verify, evaluate, incentivize, and support research. The purpose of this study is to identify how researchers are studied as research subjects. We are thus using the method of scoping review to conduct this meta-research study. 

3. Motivating the research question: In my view there is no sufficient derivation of the research question from the research literature (e.g., quantitative scientific studies) on questions where the researchers themselves were made the subject of the research. Reference is made to a "body of literature", but no literature is cited, although a body of empirical literature must be assumed. The question remains as to the scientific background against which the results in the registered report can be interpreted and used.

We appreciate the importance of presenting the body of literature to the reader, however there are no readily citable references that encapsulate, summarize, or discuss as a whole (or in part) the body of literature where researchers themselves are the study subjects. A citation to this literature would simply be a collection of example studies. This gap in the literature is specifically being addressed by the scoping review proposed in this protocol. 

Understanding this limitation, to make the statements alluding to the “body of literature” better grounded, we have revised the text to discuss the utility of meta-research more generally, citing the literature in that area and calling out research with researchers themselves as research subjects as an example. Specifically, we now state: 

(p.4, ln 88-93) Just as the examination of research practices in other areas of meta-research can help disseminate efficient and effective research policies and to identify and abandon wasteful ones, insights from the body of literature with researchers as the study subjects can illuminate how to better support researchers, enhance the development of research capacity, and advance the methods used to engage and study this population.

4. Registered report protocol: I wonder why this study was submitted as a registered report protocol. Registered report protocols are useful if you have formulated a clear research question, hypotheses, and an experimental design to test the hypotheses. This makes it possible to publish results in the subsequent study that contradict the original hypotheses. In the study plan presented no hypotheses are formulated; it is an exploratory study. A potential risk of bias is not recognisable. The plan for the data analysis and reporting is vague (e.g., tabular reports, thematic analyses, narrative summaries). Results of a preliminary study with a bibliographic database are not reported (e.g., potential number of studies, text-mining for key words). A sufficient justification for the type of article is missing.

We have submitted this scoping review protocol as a Registered Report Protocol because it is a protocol for a proposed study for which data has not yet been collected (except for testing of the proposed search strategy). As described in the PLOS One submission guidelines, a Registered Report Protocol is: “an article describing the study design, rationale, timeline, proposed methodology for data collection and analysis, and where applicable ethical approval for the work. Registered Report Protocols report the study proposal prior to conducting experiments, data collection patient recruitment, and they undergo peer review to ensure that the planned research will meet PLOS ONE’s publication criteria.” This scoping review protocol falls within the purview of the Registered Report Protocol. However, because PLOS One has published scoping review protocols both as Registered Report Protocols and as Study Protocols, we can submit the protocol as a Study Protocol instead if the reviewer thinks that is more appropriate than Registered Report Protocol. 

5. Component B: It was stated (p. 9) that “These names are then combined with a series of keywords identifying actions performed by researchers as study subjects”. I wonder where these keywords (e.g., engaged, …) come from. Important verbs as for example “investigate” or “conduct” are missing. The revision should give some reasons for the selection of keywords for actions.

The intention of these key terms is to identify ways in which researchers tend to be described in the context of being themselves the subjects of research. The keywords were identified through listing terms commonly found in the titles and abstracts of known studies that met eligibility criteria for the scoping review. “Investigate” and “conduct” were not selected to be used as they do not specifically identify ways in which researchers tend to be described in the context of being themselves the subjects of research.

6. Funding: The financial disclosure stated that "funding was received from the National Institute of Aging (NIA)". This raises the question of who will fund the subsequent study.

The conduct of the scoping review will be funded with the same funding.

---

## [Decision Letter · Decision Letter 2]

5 May 2024

Identifying meta-research with researchers as study subjects: protocol for a scoping review

PONE-D-23-25128R2

Dear Dr. Laynor,

We’re pleased to inform you that your manuscript has been judged scientifically suitable for publication and will be formally accepted for publication once it meets all outstanding technical requirements.

Kind regards,

Lutz Bornmann

Academic Editor

PLOS ONE

Additional Editor Comments (optional):

Reviewers' comments:

Reviewer's Responses to Questions

**Comments to the Author**

1. Does the manuscript provide a valid rationale for the proposed study, with clearly identified and justified research questions?

Reviewer #2: Yes

2. Is the protocol technically sound and planned in a manner that will lead to a meaningful outcome and allow testing the stated hypotheses?

Reviewer #2: Yes

3. Is the methodology feasible and described in sufficient detail to allow the work to be replicable?

Reviewer #2: Yes

4. Have the authors described where all data underlying the findings will be made available when the study is complete?

Reviewer #2: Yes

5. Is the manuscript presented in an intelligible fashion and written in standard English?

Reviewer #2: Yes

6. Review Comments to the Author

You may also provide optional suggestions and comments to authors that they might find helpful in planning their study.

Reviewer #2: All the reviewer's comments have been taken into account in the revision process. I very much appreciate the changes that have been made to improve the manuscript. I look forward to the publication and, especially, to the later results of the project.

7. PLOS authors have the option to publish the peer review history of their article (what does this mean?). If published, this will include your full peer review and any attached files.

Reviewer #2: No

---

## [Editor Report · Acceptance letter]

8 May 2024

PONE-D-23-25128R2 

PLOS ONE

Dear Dr. Laynor, 

I'm pleased to inform you that your manuscript has been deemed suitable for publication in PLOS ONE. Congratulations! Your manuscript is now being handed over to our production team.

Kind regards, 

on behalf of

Dr. Lutz Bornmann 

Academic Editor

PLOS ONE